# Golgi Complex: A Signaling Hub in Cancer

**DOI:** 10.3390/cells11131990

**Published:** 2022-06-21

**Authors:** Daniela Spano, Antonino Colanzi

**Affiliations:** 1Institute of Biochemistry and Cell Biology, National Research Council, Via Pietro Castellino 111, 80131 Naples, Italy; 2Institute for Endocrinology and Experimental Oncology “G. Salvatore”, National Research Council, 80131 Naples, Italy; a.colanzi@ieos.cnr.it

**Keywords:** Golgi Complex, cancer, signaling pathways, cancer hallmarks

## Abstract

The Golgi Complex is the central hub in the endomembrane system and serves not only as a biosynthetic and processing center but also as a trafficking and sorting station for glycoproteins and lipids. In addition, it is an active signaling hub involved in the regulation of multiple cellular processes, including cell polarity, motility, growth, autophagy, apoptosis, inflammation, DNA repair and stress responses. As such, the dysregulation of the Golgi Complex-centered signaling cascades contributes to the onset of several pathological conditions, including cancer. This review summarizes the current knowledge on the signaling pathways regulated by the Golgi Complex and implicated in promoting cancer hallmarks and tumor progression.

## 1. Introduction

In mammals, the Golgi Complex (GC) consists of a continuous membranous system composed of stacks connected by tubular bridges, thus forming a structure known as the “Golgi ribbon”. This organelle plays a central role in the trafficking, processing and sorting of membranes, proteins and lipids. In addition to these classical functions, several studies provide evidence that the GC contributes to the regulation of many cellular processes, such as migration, mitosis, apoptosis, inflammation, DNA repair, autophagy and stress responses [1]. GC scaffold proteins (including golgins, GRASPs and PAQRs) and GC-localized signaling molecules (such as phosphoinositides, small GTPases, kinases and phosphatases [2,3,4]) mediate these processes. Hence, the GC is now recognized as a hub where distinct signaling pathways originate for the control of cellular processes both in healthy and pathological cells [5].

Cancer cells show functional and structural GC disorganization, which has been associated with cancer development and progression. Aberrant glycosylation [6,7], abnormal expression of Ras GTPase and Rab over-activation [8,9,10], dysregulation of kinases [11], hyperactivation of myosin motor proteins [12], altered sialylation [13] and enhanced trafficking, modified expression of GC proteins [7,14] are common aspects of cancer. There are several excellent reviews on the GC functional dysregulation in cancer and the roles played by selected GC proteins in cytophysiology and cancer development and progression, as well as the correlation between their expression and cancer clinicopathological features [6,7,11,13,14,15]. However, an updated review comprising the roles played by GC and GC-localized proteins in modulating cancer-related signaling pathways is lacking. Thus, this review focuses on the GC-centered signaling pathways involved in cancer development and progression.

## 2. GC-Centered Signaling Pathways Regulate Cancer Hallmarks

Cancer cells are characterized by genomic instability, sustained proliferation, increased migration and invasion, resistance to apoptosis, ability to promote angiogenesis, evasion from immune destruction, resistance to cancer conventional therapies and cancer metabolism reprogramming [16]. Accumulating evidence shows the involvement of several GC-localized proteins in promoting cancer hallmarks. One of the more relevant examples is Golgi phosphoprotein 3 (GOLPH3), whose functions in cancer progression and signaling pathways have been fully reviewed by Sechi and collaborators [17]. Although GOLPH3 will not be further tackled in this review, it is relevant to note that a recent study unravels an additional molecular mechanism through which GOLPH3 contributes to tumorigenesis. The authors identify GOLPH3 as a master regulator of glycosphingolipid metabolism and show that in cancer, the increased GOLPH3 levels alter glycosphingolipid synthesis and plasma membrane composition, thus promoting mitogenic signaling and cell proliferation [18]. Although the involvement of GC and its proteins in promoting cancer hallmarks has been widely demonstrated, the GC-centered signaling cascades underlying these hallmarks have not been fully revealed. This review describes the current knowledge on GC-centered signaling pathways that regulate the main cancer hallmarks. For an immediate overview of the GC-centered signaling cascades involved in cancer progression, the reader is referred to Table 1 and Figure 1.

### 2.1. GC-Centered Signaling Pathways that Regulate Cancer Migration, Invasion and Metastasis Formation

#### 2.1.1. ADP-Ribosylation Factor 1 (ARF1)

ADP-ribosylation factor 1 (ARF1) is a GC-localized small GTP-binding protein. In the GTP-bound form, ARF1 promotes the recruitment at the GC of a variety of effectors, including coat proteins involved in vesicle formation (e.g., COPI), lipid-modifying enzymes and the golgins GMAP-210 and Golgin160 [108]. Therefore, ARF1 is involved in vesicular trafficking, lipid homeostasis and organelle dynamics. For example, Arf1 controls the intra-GC and GC-to-endoplasmic reticulum (ER) transport of cargo proteins through recruiting to the GC the coat proteins of COPI carriers. Moreover, Arf1 binds to and promotes PKD recruitment to the trans-Golgi network (TGN), which is required for membrane fission to generate cell surface-specific transport carriers. ARF1 is associated with the plasma membrane in some cell types and cycles off the GC to the cytosol upon specific conditions [109,110]. Accumulating evidence suggests that ARF1 plays a role in the migration and invasion of cancer cells. In non-stimulated, highly invasive breast cancer cells, ARF1 is partially localized to dynamic plasma membrane ruffles where epidermal growth factor (EGF) stimulation promotes its rapid and transient activation. EGF-activated ARF1 triggers PI3K/AKT cascade by inducing both the recruitment of the PI3K catalytic subunit p110α and AKT to the plasma membrane and the interaction between the activated EGF receptor (EGFR) and the PI3K regulatory subunit p85α [19]. In turn, the PI3K/AKT pathway activation promotes cell migration and proliferation. In addition, in invasive breast cancer cells, ARF1 constitutively binds Rac1, a Rho GTPase associated with lamellipodia formation during cell migration [21]. Upon EGF stimulation, ARF1 induces the GTP-loading of Rac1, Rac1 association with its effector IRSp53 and the translocation of both Rac1 and IRSp53 proteins to the plasma membrane, a key event for lamellipodia formation and cell migration [21]. Another mechanism by which ARF1 regulates the migration of highly invasive breast cancer cells consists of controlling an EGF-dependent assembly of focal adhesions [22]. Here, EGF stimulates ARF1 to form a complex with the key focal adhesion proteins, including paxillin, talin, β1-integrin and focal adhesion kinase (FAK). Upon EGF treatment, ARF1 induces the interaction of paxillin, talin and FAK with each other and their recruitment to β1-integrin at focal adhesion sites. Then, ARF1 promotes the EGF-induced phosphorylation and activation of FAK and Src; subsequently, ARF1-activated Src further phosphorylates FAK. These phosphorylations are key events in stimulating the interaction of FAK with β1-integrin to induce adhesion [22]. Conversely, FAK modulates the EGFR phosphorylation levels, thus resulting in ARF1 activation [22]. Schlienger S and collaborators unravel an additional mechanism by which ARF1 supports the invasion of breast cancer cells [23]. They show that ARF1 stimulates the maturation of invadopodia, the release of shedding microvesicles and the activity of matrix metalloproteinase-9 (MMP-9), which promotes the degradation of the extracellular matrix (ECM) and, consequently, cancer cells invasion [23]. From a molecular point of view, upon EGF treatment, ARF1 interacts with and activates RhoA and RhoC GTPases, which, in turn, phosphorylate myosin light chain (MLC) thus inducing the contraction of the actin-based cytoskeleton and generating the force required for microvesicle fission. In addition, a positive feedback regulatory mechanism exists in which activated RhoA and RhoC further support the activation of ARF1 [23]. Recent studies demonstrate that the activation of ARF1, specifically at the GC, directly modulates prostate cancer cell migration and invasion in response to the activation of G protein-coupled receptor (GPCR) CXCR4. From a molecular point of view, the stromal cell-derived factor 1α (SDF1α)-mediated activation of plasma membrane CXCR4 induces the translocation of Gβγ dimers from the plasma membrane to the GC. Here, Gβγ dimers trigger PI3Kγ, which, in turn, promotes the activation and the recruitment to the GC of ARF1. Then, ARF1 stimulates ERK1/2, thus inducing prostate cancer cell migration and invasion [24,111]. Therefore, GC serves as a platform to compartmentalize the events involved in GPCR signaling, including the translocation of Gβγ and sequential activation of PI3Kγ, ARF1 and MAPK pathways. Recent evidence shows that ARF1 stimulates epithelial ovarian cancer cell migration by interacting with PI3K and activating its signaling cascade [20]. Moreover, ARF1 is also a PI3K downstream target, thus suggesting that ARF1 forms a PI3K-dependent feedforward signaling pathway [20].

#### 2.1.2. GM130

GM130 is a cis-Golgi matrix protein involved in the maintenance of the GC structure and in the stacking of GC cisternae. GM130 is supposed to play an indirect role in the GC structure maintenance by the tethering interaction required for membrane traffic. In addition, it plays a role in recruiting protein complexes involved in microtubule polymerization and polarity-based signaling in a large variety of cell functions [112]. For example, GM130 forms a protein complex with AKAP450, CEP215 and MMG, thus recruiting γ-TuRC for GC-dependent microtubule nucleation. GM130 activates the kinase YSK1 and the exchange factors TUBA and RasGFR to modulate cell polarity. In addition, the GM130/TPX2/Aurora-A pathway controls spindle formation and orientation [113]. The role of GM130 in tumorigenesis depends on the type of cancer. GM130 is strongly up-regulated in gastric cancer and contributes to the gastric cancer cell migration and invasion by stimulating the transcription of the Snail gene, thus inducing the epithelial to mesenchymal transition (EMT) characterized by decreased expression of E-cadherin epithelial marker and increased expression of N-cadherin and vimentin mesenchymal markers [26]. Conversely, GM130 is frequently down-regulated in colorectal and breast cancer, where it modulates cell migration and invasion through the regulation of the activity of the Cdc42 GC-localized pool [27,28]. At GC, GM130 interacts with RasGRF2, a member of the RasGRF family of Guanine Nucleotide Exchange Factors, which is a repressor of Cdc42, a key regulator of cell polarity, and an activator of Ras [114]. This interaction prevents RasGRF2 from binding and inhibiting Cdc42. The GM130 loss causes the release of RasGRF2, which is free to bind to GC-localized Cdc42 and inhibit its activity [27], thus resulting in the loss of the asymmetric front–rear Cdc42-GTP distribution in directionally migrating cells and cell polarity and in the reduction of E-chaderin expression, which is associated with increased metastatic potential [115]. In addition, the release of RasGRF2 activates the Ras-ERK1/2 signaling pathway [27]. According to this molecular mechanism of action, GM130 loss inhibits directed breast cancer cells motility while increasing random cell motility, cell velocity and cell invasion, thus promoting tumor progression [28].

#### 2.1.3. Golgi Membrane Protein 1 (GOLM1)/Golgi Protein 73 (GP73)/Golgi Phosphoprotein 2 (GOLPH2)

Golgi membrane protein 1 (GOLM1) is a 73-kDa type II cis- and medial-Golgi-localized transmembrane glycoprotein, also called Golgi protein 73 (GP73) or Golgi phosphoprotein 2 (GOLPH2). GOLM1 is expressed in both normal and tumoral human tissues of the epithelial lineage and plays a key role in the sorting and modification of cargo proteins in the ER and protein transportation through the GC. Its expression is up-regulated in several cancer types, including hepatocellular carcinoma (HCC), glioma and bladder, lung, prostate and oesophageal cancers [29,31,33,34,35,36,37,38,116]. It acts as an oncogene by inducing cancer cell proliferation, migration and invasion and impairing the chemotherapeutic agents-induced apoptosis by modulating several signaling pathways [29,31,33,34,35,37,38,116]. GOLM1 stimulates HCC invasion and metastasis by multiple signaling cascades. GOLM1 enhances the expression of cAMP responsive element-binding protein (CREB) transcription factor, which, in turn, activates the expression of the MMP13 gene, thus promoting HCC invasion and metastasis [29]. In turn, MMP-13 increases GOLM1 expression in HCC cells, thus suggesting the existence of a positive feedback loop between the two proteins [29]. Another molecular mechanism by which GOLM1 promotes HCC invasion is the regulation of MMP2 intracellular trafficking and secretion [30]. Herein, GOLM1 directly interacts with intracellular MMP-2 through its cytoplasmic domain, then both proteins translocate to the plasma membrane and are secreted, which, consequently, stimulates cell invasion [30]. The inhibition of MMP-2 trafficking through GOLM1 silencing drives the accumulation of intracellular MMP-2, which binds Src. The MMP-2/Src interaction inhibits the phosphorylation of Src at Y416, thus resulting in the inhibition of phosphorylation and the nuclear translocation of p-JNK1/2 (T183/Y185). The inhibition of p-JNK1/2 (T183/Y185) nuclear translocation causes the impairment of the p53-p21 signaling pathway and the inactivation of the pRb (S780) phosphorylation, thus promoting the Rb-E2F1 complex formation, which reduces the content of free transcription factor E2F1, hence resulting in the inhibition of E2F1 target genes transcription, including MMP-2 [30]. Therefore, GOLM1 participates not only in the trafficking and secretion of MMP-2 but also in regulating MMP-2 transcription by activating a negative feedback loop. GOLM1 drives HCC metastasis by interacting with EGFR/receptor tyrosine kinase (RTK) [31]. Upon EGF stimulation, GOLM1 can transfer from TGN to the cytosol and form a complex with Rab11 GTPase and EGFR/RTK once it is internalized. This interaction assists the EGFR/RTK anchoring on TGN and EGFR/RTK polarized recycling back to the plasma membrane towards migration direction. This event drives the sustained activation of the EGFR downstream signaling effectors, including AKT and S6 kinase (S6K), which culminates in the increased expression of MMP9 and the reduced expression of E-cadherin, thus leading to cancer invasion and metastasis [31]. Gai X and collaborators demonstrated that exosomes secreted from HCC cell lines contain a secreted form of GOLM1, which induces the activation of GSK3β and the expression of MMP1 and MMP9 thus resulting in the migration and invasion of HCC recipient cells [32]. In the same line of evidence, GOLM1 stimulates prostate cancer migration and invasion by triggering the PI3K/AKT/mTOR signaling pathway [33]. Similarly, GOLM1 promotes PDGFA/PDGFRα-mediated migration and invasion of glioma through the activation of PI3K/AKT/mTOR cascade, which, in turn, induces the activation of GSK3β and the increased expression of ZEB1 and Snail [34]. In addition, a positive feedback loop exists in which the mammalian target of rapamycin complex 1 (mTORC1) stimulates the expression of GOLM1 by suppressing the expression of miR-145, a negative regulator of GOLM1 expression [32,39]. 

Apart from the regulation of the above-mentioned signaling pathways, GOLM1 participates in other signaling pathways that promote migration and invasion. GOLM1 facilitates the TGF-β1-induced EMT and invasion in HCC and bladder cancer [35,36]. TGF-β1 stimulation enhances p-Smad2 and p-Smad3 levels, increases the expression of vimentin and reduces the expression of E-cadherin, thus inducing EMT and cell invasion. GOLM1 overexpression further increases p-Smad2 and p-Smad3 levels, EMT and cell invasion, thus indicating that GOLM1 strengthens the canonical TGF-β1/Smad2/Smad3 signaling [35,36]. 

GOLM1 modulates glioblastoma cell migration, invasion and EMT by regulating the Wnt/β-catenin signaling cascade [37]. GOLM1 silencing results in a reduced level of p-GSK3β (Ser9) inhibitory phosphorylation, which leads to increased GSK3β activity and, in turn, to reduced β-catenin level and decreased β-catenin translocation to the nucleus, thus suggesting that GOLM1 positively modulates the Wnt signaling pathway. The Wnt signaling impairment causes the decreased expression of EMT-related markers, such as Snail and MMP2, hence resulting in the inhibition of glioblastoma cell migration and invasion [37]. 

Recently, Song Q and collaborators employed a global phosphoproteomics approach to acquire further insights into the signaling pathways regulated by GOLM1 responsible for promoting lung cancer malignancy [38]. GOLM1 overexpression induces the expression of genes enriched in the MAPK signaling pathway. Among the differentially expressed genes in the MAPK signaling cascade, p53 is located at the central position of all the hub genes and ranked first of them. GOLM1 overexpression enhances the phosphorylation of p53 protein at S315, which inactivates p53 by increasing its degradation [117]. Therefore, GOLM1 overexpression promotes lung cancer malignant progression by reducing the p53 stability, which results in weakening the p53-mediated inhibition of tumor formation [38]. 

#### 2.1.4. Similar Expression to FGF (Sef) (also Known as Interleukin-17 Receptor D (IL-17RD))

Similar expression to FGF (Sef) is predominantly localized at GC and acts as a scaffold for the assembly of several receptor complexes and their interacting proteins to generate signaling outputs. It is an inhibitor of RTKs signaling, thus suggesting a role for Sef as a tumor suppressor [45]. Sef expression is strongly reduced in several cancer types, including breast, thyroid, ovarian, cervical and advanced prostate cancers [41,45,118]. Sef knockdown facilitates a more invasive phenotype and enhances the fibroblast growth factor (FGF) 8-induced migration and the invasion of prostate cancer cells by stimulating the MAPK signaling pathway, which results in increased MMP9 expression [41]. Conversely, increased Sef expression significantly impairs both in vitro prostate cancer cell migration and invasion and in vivo prostate xenograft metastases [42,43]. Sef blocks multiple FGF-induced signaling by reducing the intensity and duration of ERK phosphorylation, which causes the impairment of nuclear translocation and transcriptional activity of ERK. The transfection of constitutively active Ras overcomes the Sef inhibitory effects on prostate cancer cells invasion, thus suggesting that the point of Sef action is likely either at the level of the FGF receptor or at the level of Ras [42]. Therefore, the loss of Sef expression results in unattenuated FGF signaling, which leads to prostate cancer progression and metastasis [41,42]. To further unravel the molecular mechanisms by which Sef impairs prostate cancer metastasis, Hori S and collaborators performed phosphokinase arrays demonstrating that Sef attenuates the signaling of not only ERK-MAPK but also JNK and p38 pathways as well, all involved in mediating EMT [43]. The impairment of these signaling cascades results in the altered expression of EMT genes with E-chaderin being up-regulated and Versican, SIP1, ZEB2, WNT5B, ITGA5, IGFBP4, STEAP1 and SNAI2 being suppressed [43]. Similarly, Sef impairs EMT and the acquisition of metastatic phenotype both in in vitro and in vivo breast cancer [44]. Herein, Sef interacts with β-catenin and causes increased membrane and cytosolic accumulation of β-catenin, and, consequently, reduced nuclear localization and transcriptionally active form of β-catenin, thus resulting in increased expression of E-cadherin epithelial marker and decreased expression of EMT markers, including Snail, Slug, ZEB1 and N-Cadherin [44]. 

#### 2.1.5. Golgin-97

Golgin-97, a coiled-coil protein localized at TGN, acts as a tethering molecule involved in vesicular trafficking of a specific class of basolateral cargoes (such as E-cadherin) and the maintenance of cell polarity. During traffic, the protein kinase D (PKD) is activated at the TGN and phosphorylates the mono-ADP-ribosyltransferase PARP12, which, in turn, mono-ADP-ribosylates Golgin-97, which mediates basolateral cargoes (including E-cadherin) export and carrier fission, thus contributing to the maintenance of E-cadherin-mediated cell polarity and cell–cell junctions [119]. Golgin-97 low expression is correlated with breast cancer invasiveness and poor overall survival of cancer patients, thus suggesting that Golgin-97 is a tumor suppressor that inhibits cancer invasiveness. Its down-regulation in breast cancer cells induces a reduction of IκBα levels, which results in the activation of nuclear factor kappa B (NF-κB), its nuclear translocation and, in turn, the expression of its target genes, which promote cell migration and invasion [51]. The molecular mechanism by which Golgin-97 regulates IκBα is still unknown. Golgin-97 does not interact with the IκB kinase β (IKKβ), whereas it probably interacts with some unidentified molecule(s) to modulate IκBα levels, thus inhibiting NF-κB activation.

#### 2.1.6. TMED Family of p24 Proteins

The members of the transmembrane emp24 domain-containing TMED family of p24 proteins play roles in bidirectional vesicular cargo trafficking from the ER to the GC [120]. As such, the altered expression and/or function of these proteins may affect the transport of proteins in the secretory pathway, thus contributing to multiple diseases, including cancer.

TMED2 regulates the transport of cargo proteins; therefore, its abnormal expression may cause uncontrolled protein transport. It promotes the migration and invasion of ovarian cancer cells by activating the insulin-like growth factor (IGF) 2/IGF receptor (IGF1R)/PI3K/AKT signaling pathway by two molecular mechanisms [52]. On the one side, TMED2 directly binds to AKT2, thereby facilitating its phosphorylation and, consequently, its activation; on the other side, TMED2 mRNA serves as a competing endogenous RNA to regulate the expression of IGF1R through competing for miR-30a. Therefore, TMED2 mRNA binding to miR-30a prevents the miR-30a binding to IGF1R mRNA, thus increasing the IGF1R expression and, in turn, the IGF2/IGF1R/PI3K/AKT signaling cascade [52]. 

TMED3 plays a role in the selection and secretion of COP vesicles in the ER-GC network and modulates several signaling pathways. TMED3 expression is strongly up-regulated in several cancer types (including osteosarcoma, breast cancer, chordoma, HCC and non-small cell lung cancer (NSCLC)) and correlates with poor prognosis in patients [54,55,56,57,121]. TMED3 function is required for WNT ligands normal intracellular localization, trafficking and secretion [122,123]. Indeed, TMED3 knockdown causes the accumulation of WNT ligands, mainly at ER, and to a lesser extension at the GC, and impairs their secretion, thus resulting in the suppression of canonical WNT-TCF signaling, as exemplified by the drastic reduction in the expression of TCF target genes (such as AXIN2, EPHB2, SOX4 and P21) [53]. In colon cancer, endogenous WNT signaling simultaneously promotes primary tumorigenesis and prevents metastasis [124]. Therefore, TMED3, being a positive regulator of canonical WNT-TCF signaling, acts as a suppressor of colon cancer metastases [53]. TMED3 knockdown results in the increased expression of TMED9, another member of the p24 proteins family involved in cargo selection in the processing ER-GC network of proteins and innate immune signaling. TMED9 antagonizes TMED3 function through promoting colon cancer metastases [58]. Indeed, TMED9 knockdown enhances the expression of metastatic suppressor genes (such as AKAP12) and genes coding WNT signaling components (including WNT11, WNT3, MUC16, VGLL1, SOSTDC1 and LGR5). Moreover, TMED9 silencing reduces the expression of genes involved in EMT (including MMP28, ADAM8 and SNAI3) and cancer progression (such as DPEP1, LAMP3 and GSPG4). Interestingly, the opposite gene regulation is found between TMED3 silencing and TMED9 silencing, thus suggesting that these two TMED proteins have antagonistic actions in regulating the expression of multiple genes involved in the metastasis process [58]. Among the genes repressed by TMED9 silencing, CNIH4, PIGA, SMIM13 and C11orf24 genes, encoding proteins localized in the secretory network, are identified. In particular, CNIH4 is a member of the CORNICHON family of TGFα exporters and is required for TGFα trafficking, membrane localization and secretion. Once secreted, TGFα activates its receptor EGFR, thus stimulating the ERK, AKT and Hedgehog (HH)-GLI signaling, which culminates in colon cancer cells migration and metastasis [58]. In TMED9 silenced cells, TGFα appeared retained in the GC, which compromises its function, thus resulting in the suppression of colon cancer cells migration and metastasis. In summary, TMED3 positively modulates WNT-TCF signaling, which suppresses the metastatic potential of colon cancer cells. Moreover, WNT-TCF inhibits TMED9, which, in turn, represses WNT-TCF pathway components and drives CNIH4/TGFα/GLI signaling, thus promoting colon cancer metastases. Therefore, the metastatic transition in colon cancer is caused by a pathway switch in which WNT-TCF signaling is suppressed, and the HH-GLI1 pathway is enhanced [124]. TMED9 and TMED3 play a role in this metastatic transition with their activities that balance each other to determine metastatic outcomes of colon cancer cells and control, in opposite manners, a global gene cohort that includes multiple factors implicated in the regulation of metastases [53,58]. 

Unlike the above-presented data, TMED3 serves as a promoter of in vitro HCC cell migration and invasion and in vivo metastases through enhancing IL-11 expression [54]. The TMED3-mediated IL-11 increased expression causes the IL-11-enhanced secretion, which, in turn, stimulates the phosphorylation of Signal transducer and activator of transcription 3 (STAT3), thus suggesting that TMED3 promotes HCC metastases through activating the IL-11/STAT3 signaling pathway [54]. Similarly, TMED3 promotes breast cancer cell migration and invasion by activating the Wnt/β-catenin signaling cascade [55]. TMED3 overexpression induces the substantial accumulation of β-catenin in the cytoplasm and nucleus and Axin2 in the cytoplasm, thus resulting in the increased expression of MMP2, MMP7 and MMP9 target genes [55]. The TMED3-mediated activation of Wnt/β-catenin signaling is further validated by Zhang D and collaborators, who demonstrate that in NSCLC, TMED3 stimulates the activation of AKT, which, in turn, phosphorylates GSK3β at Ser9, causing GSK3β inactivation, which, in turn, leads to β-catenin activation, thus resulting in increased expression of N-cadherin and vimentin, decreased expression of E-cadherin, and, consequently, increased invasion of NSCLC cells [56]. 

TMED10 regulates vesicular protein trafficking serving as a cargo receptor. Recently, TMED10 has been identified as a key player in the unconventional secretion of cytosolic proteins lacking a secretion signal peptide (called leaderless cargoes). In this context, TMED10 acts as a protein channel for the vesicle entry and secretion of many leaderless cargoes [125]. TMED10 inhibits the TGF-β-induced migration of lung cancer cells, thus serving as a tumor suppressor [59]. TMED10 binds to both TGF-β type I (also termed ALK5) and type II receptors (TβRII) and disrupts the TGF-β-induced heteromeric complex formation, thus impairing the phosphorylation of Smad2 and the Smad-dependent transcriptional activity. In addition, since the TGF-β receptor complex can also signal through a non-Smad pathway, including JNK and p38 [126], TMED10 is able to suppress the TGF-β-mediated activation of JNK and p38 pathways as well [59]. 

#### 2.1.7. Secretory Carrier-Associated Membrane Protein 1 (SCAMP1)

SCAMP1, a protein involved in post-Golgi recycling pathways and endosome cell membrane recycling, enhances the transport of metastasis suppressor protein 1 (MTSS1) to the plasma membrane. MTSS1, a member of the IMD-family (IRSp53 and MIM (missing in metastasis) domain), serves as an actin-binding scaffold protein and stimulates the activation of Rac1-GTP, thus promoting cell–cell adhesions and preventing HER2^+^/ER^−^/PR^−^ breast cancer cell migration and invasion [62]. Therefore, these findings suggest a role for SCAMP1 in preventing HER2^+^/ER^−^/PR^−^ breast cancer invasion through stimulating the MTSS1/Rac1-GTP axis [62].

#### 2.1.8. RKTG (Raf Kinase Trapping to Golgi)/PAQR3

The Raf kinase trapping to Golgi (RKTG) protein, also called PAQR3, belonging to the progestin and adipoQ receptor (PAQR) family, is a GC-anchored membrane protein. At the GC, RKTG/PAQR3 promotes Gβγ-mediated activation of PKD, which, in turn, stimulates the fission of GC transport vesicles directed towards the plasma membrane [127]. Its expression is reduced in several cancer types (including breast cancer, prostate cancer, glioma, esophageal squamous cell carcinoma, laryngeal squamous cell carcinoma, gastric cancer, NSCLC and colorectal cancer) [65,66,67,69,74,75,128] due to either the hypermethylation of PAQR3 gene promoter [64,129], the increased expression of miRNA targeting PAQR3 [130] or the DDB2-mediated ubiquitination and degradation [131]. RKTG/PAQR3 expression inversely correlates with cancer malignancy and poor prognosis [65,66,67,74,75,128], thus suggesting that RKTG/PAQR3 acts as a tumor suppressor. RKTG/PAQR3 binds to and translocates Raf-1 to the GC, thus inhibiting Raf-1 activation and its interaction with Ras and MEK, which, consequently, results in the suppression of Ras/Raf/MEK/ERK signaling pathway activation [132]. RKTG/PAQR3-mediated inhibition of Raf/MEK/ERK cascade (exemplified by decreased protein expression levels of Raf-1, p-MEK1 and p-ERK1/2) impairs EMT and, consequently, in vitro cell migration and invasion of esophageal cancer cells and laryngeal squamous cell carcinoma cells [63,64,65,66]. RKTG/PAQR3 suppresses EMT phenotype, migration and invasion of multiple cancer cells (including gastric cancer, prostate cancer and glioma cells) by inhibiting not only Ras/Raf/MEK/ERK signaling but also the PI3K/AKT pathway by trapping to the GC key players in these cascades [67,68,69]. As previously described, ERK signaling inhibition is mediated by sequestering Raf-1 to the GC, while AKT cascade suppression is mediated by trapping the Gβ subunit and p110α subunit of PI3K to the GC. In detail, RKTG/PAQR3 inhibits Gβ/γ subunit-mediated activation of AKT upon GPCR activation by retaining the Gβ subunit to the GC [133]. Moreover, RKTG/PAQR3 suppresses PI3K activation and AKT phosphorylation by interacting with and sequestering the p110α subunit of the PI3K complex to the GC and, consequently, impairing its interaction with the p85 regulatory subunit [134]. 

Another molecular mechanism by which RKTG/PAQR3 suppresses EMT is through modulating Twist1 protein stability and degradation [70]. RKTG/PAQR3 forms a protein complex with Twist1 and BTRC, the E3 ubiquitin ligase of Twist1, hence enhancing the interaction between Twist1 and BTRC, which promotes BTRC-mediated Twist1 polyubiquitination, its translocation from the nucleus to the proteasome-containing structure in the cytoplasm and its degradation, thus leading to the suppression of both gastric cancer cells in vitro EMT phenotype and migration and in vivo metastases [70]. 

#### 2.1.9. Protein Kinase D (PKD) Family

The PKD family of serine/threonine protein kinases belongs to the calcium-/calmodulin-dependent protein kinase superfamily and consists of three members: PKD1, PKD2 and PKD3. PKD proteins localize at several subcellular compartments, including cytoplasm, plasma membrane, GC, mitochondrion, ER and the nucleus. Among these multiple subcellular localizations, the GC and plasma membrane represents the main PKD protein localizations. At the GC, they modulate several cellular processes, such as the fission of protein and lipid cargo vesicles from the TGN to the plasma membrane, cell shape, movement and invasion [135,136]. PKD’s dysregulation is associated with several pathological conditions, including cancer, where the three PKD family members may have different functions [137]. In consideration of the relevant roles played by PKDs in regulating cancer-related signaling pathways, these proteins will be tackled in this review, although, to date, the precise subcellular localization from which the PKD-regulated signals emanate has not yet been defined.

PKD1 was found down-regulated in invasive breast, advanced prostate and gastric cancers with low expression associated with cancer aggressiveness and metastasis [89,138,139,140], thus suggesting that this kinase serves as a tumor suppressor. PKD1 negatively modulates the Wnt/β-catenin signaling pathway through its ability to phosphorylate β-catenin at T120, which results in β-catenin localization at TGN and the inhibition of its transcriptional activity [88]. Therefore, PKD1 down-regulation contributes to cancer development and progression by stimulating the Wnt/β-catenin signaling cascade [141].

Another mechanism through which PKD1 suppresses cancer aggressiveness consists of the inhibition of EMT. In prostate cancer, PKD1 binds to and phosphorylates the transcription factor Snail on Ser11, thus creating a binding site for 14-3-3 proteins, which interact with Snail and promote its nuclear export. Consequently, the Snail transcriptional activity is inhibited, thus resulting in the induction of E-cadherin expression and the inhibition of N-cadherin and vimentin mesenchymal markers expression, which causes the impairment of EMT [89]. In addition, PKD1 colocalizes with E-cadherin at the cell junctions, and binds to and phosphorylates E-cadherin [90], thus stabilizing the interaction of E-cadherin with catenins, which induces cell–cell adhesion and reduces prostate cancer cells motility [90]. Therefore, PKD1 down-regulation in prostate cancer further activates the Wnt/β-catenin signaling pathway by destabilizing the E-cadherin/β-catenin complex, which leads to increased amounts of β-catenin available for translocation to the nucleus. 

PKD2 is highly expressed in human HCC, where it contributes to tumor necrosis factor-alpha (TNFα)-induced EMT and metastasis [99]. The binding of TNFα to TNF-receptor-type 1 (TNFR1) triggers the interaction of TNFR1 with TNFR-associated factor 2 (TRAF2), which activates protein kinase C δ (PKCδ), which, in turn, activates PKD2. Active PKD2 binds to the p110α and p85 subunits of PI3K and stimulates the PI3K/AKT signaling pathway, which, in turn, phosphorylates GSK3β on its inhibitory phosphorylation sites, thus inducing the accumulation of β-catenin in the nucleus, which stimulates the expression of N-cadherin and vimentin mesenchymal markers and suppresses the expression of E-cadherin and ZO-1 epithelial markers, thus promoting HCC EMT and invasion [99]. 

Both PKD2 and PKD3 have a cooperative role in prostate cancer cell migration and invasion [100]. PKD2 and PKD3 promote NF-κB signaling, which stimulates the expression of invasion- and metastasis-related genes, including MMP14, urokinase-type plasminogen activator (uPA) and uPA receptor (uPAR), and reduces the expression of plasminogen activator inhibitor-2 (PAI-2). Then, the serine protease uPA-uPAR signaling activates a cascade of MMPs, which degrade ECM, thus promoting ECM remodeling and cancer cell invasion and metastasis [100]. Although both PKD2 and PKD3 are required for NF-κB-mediated transactivation, they activate this pathway differentially. Indeed, PKD2 is primarily responsible for nuclear translocation of p65 NF-κB through activating the phosphorylated IkBα kinase, a phosphorylated inhibitor of NF-κB and IkBα degradation (pIKK-pIkBα-IkBα) cascade, in which the phosphorylation of IKK leads to the phosphorylation and degradation of IkBα, thus causing the p65 nuclear translocation. On the other hand, PKD3 enhances cancer cell invasion mainly through interacting with and suppressing the constitutive expression of histone deacetylase 1 (HDAC1), which binds to the uPA promoter and negatively regulates uPA transcription. Although the exact mechanism of HDAC1 suppression by PKD3 is not yet clear, it results in the additional transcriptional activation of uPA independent of PKD2-mediated p65 translocation, thus further promoting the cell invasion and metastasis [100]. 

### 2.2. GC-Centered Signaling Pathways that Regulate Cancer Proliferation

#### 2.2.1. ADP-Ribosylation Factor 1 (ARF1)

ARF1 regulates the proliferation of breast cancer cells by modulating pRb hyperphosphorylation and its association with E2F1 [25]. In cell proliferating, ARF1 is highly activated, is mainly localized at GC, is associated with plasma membrane ruffles [19], is poorly associated with the chromatin in the nucleus and does not bind pRb [25]. In this condition, pRb is hyperphosphorylated and dissociated by the E2F1 transcription factor, which translocates into the nucleus and activates the transcription of its target genes (such as cyclin D1, Mcm6 and E2F1), thus facilitating the G1 to S transition [25]. In non-proliferating cells arrested in the G0/G1 phase, ARF1 is mainly inactive, enriched in the chromatin at E2F-responsive promoter sites and bound to pRb. pRb is hypophosphorylated and forms a complex with E2F1, thus suppressing the cell cycle progression and inducing senescence [25]. The ARF1-mediated stimulation of cancer cell proliferation is further corroborated by Gu and collaborators’ study, which provides evidence that ARF1 interacts with and activates PI3K, thus stimulating the phosphorylation of AKT. The ARF1 mediated-activation of the PI3K signaling cascade promotes the G0/G1 to S phase transition of epithelial ovarian cancer cells and, in turn, the cell proliferation [20].

#### 2.2.2. Golgi Membrane Protein 1 (GOLM1)/Golgi Protein 73 (GP73)/Golgi Phosphoprotein 2 (GOLPH2)

GOLM1 promotes HCC, glioma and prostate cancer proliferation and growth through the regulation of the EGFR/PDGFRα/RTK signaling pathway, thus resulting in the activation of PI3K/AKT/mTOR signaling cascade and in the positive feedback loop, already described in the previous section [31,32,33,34,39]. In addition, GOLM1 promotes glioblastoma cell proliferation by facilitating the Wnt signaling pathway, as described in the previous section [37]. Here, GOLM1 silencing impairs Wnt/β-catenin signaling, which, in turn, causes a decreased expression of proliferation-associated proteins (including CyclinD1, CyclinE1, c-Myc and p-AKT), thus resulting in cell cycle arrest in G1-S phase and cell proliferation impairment [37].

#### 2.2.3. Vesicle Transport Factor (USO1) (also Known as Vesicle Docking Protein, 115-KD (p115))

USO1/p115 is a member of the tether factors family involved in ER-GC trafficking and vesicular transport. USO1/p115 interacts with GM130 and Giantin and this interaction stimulates the USO1/p115 binding to Rab1, thus recruiting USO1/p115 to the COP II coated vesicles. Evidence suggests that the up-regulation of ER to GC trafficking enhances the protein transport and promotes malignant tumor progression. According to these findings, USO1/p115 silencing inhibits colon cancer cell proliferation and migration and promotes colon cancer cell apoptosis, thus suggesting a role for USO1/p115 in colon cancer progression [142]. USO1/p115 expression is up-regulated in multiple myeloma. Similarly, the USO1/p115 knockdown inhibits proliferation and induces the apoptosis of multiple myeloma cells. From a molecular point of view, USO1/p115 silencing causes reduced phosphorylation of ERK1/2 and decreased expression of proliferation-related proteins, including cyclin D1, Mcm2 and PCNA. These data suggest that USO1/p115 overexpression promotes multiple myeloma proliferation through activating the ERK1/2 signaling cascade and increasing the expression of proliferation-related factors [77].

#### 2.2.4. RKTG (Raf Kinase Trapping to Golgi)/PAQR3

RKTG/PAQR3 exerts anti-proliferative effects on multiple cancer types through negatively modulating Ras/Raf/MEK/ERK and PI3K/AKT signaling pathways by sequestering the GC key players in these cascades, as previously described in this review. In esophageal cancer, the RKTG/PAQR3-induced inhibition of the Ras/Raf/MEK/ERK pathway impairs cell cycle transition from G1 to S phase, which is associated with the induction of cell cycle inhibitors p27 and p21 and the reduction of cyclin D1, CDK4 and CDK2 [63,64]. Similarly, RKTG/PAQR3 overexpression suppresses the proliferation of laryngeal squamous cell carcinoma cells by inhibiting ERK signaling [66]. RKTG/PAQR3 overexpression in melanoma cells harboring the oncogenic mutation of B-Raf (V600E) sequestrates mutated B-Raf to the GC, thus impairing the Ras/Raf/MEK/ERK cascade and, consequently, the in vitro and in vivo melanoma cell proliferation and tumorigenicity [71]. In addition, RKTG/PAQR3-deficient mice (RKTG^-/-^) treated with chemical carcinogens show an increased proliferation rate of skin cells, shortened tumor latency and are more inclined to develop skin cancer compared to wild-type mice treated with chemical mutagens. These data indicate that RKTG/PAQR3 deficiency is able to promote the growth of chemical carcinogen-induced skin tumors. Moreover, RKTG^-/-^ mice show increased levels of phosphorylated Raf-1 and ERK both in primary keratinocytes as well as skin tumors. These findings suggest a tumor-suppressive physiological function of RKTG/PAQR3 in skin carcinogenesis via negative regulation of the Ras/Raf/MEK/ERK signaling pathway [72]. 

RKTG/PAQR3 inhibits in vitro leukemia and prostate cancer cell proliferation and in vivo prostate tumor growth by suppressing both Ras/Raf/MEK/ERK and PI3K/AKT signaling cascades [68,73]. In the same line of evidence, RKTG/PAQR3 suppresses the PI3K/AKT signaling pathway in NSCLC, thus resulting in cell cycle arrest at the G0/G1 phase, apoptosis induction and, consequently, the impairment of cell proliferation [74]. Similarly, the RKTG/PAQR3-mediated inactivation of the PI3K/AKT signaling pathway leads to the inhibition of in vitro glioma cell proliferation and attenuates in vivo xenograft tumor growth [69].

In addition to Ras/Raf/MEK/ERK and PI3K/AKT signaling cascades, RKTG/PAQR3 suppresses the Wnt signaling pathway as well. Indeed, in colorectal cancer cells, RKTG/PAQR3 overexpression inhibits both the Ras/Raf/MEK/ERK activation and the nuclear accumulation of β-catenin, thus reducing cell proliferation and colony formation [75]. In the same line of evidence, RKTG/PAQR3 depletion in the murine colorectal cancer model Apc^Min/+^, bearing the heterozygous mutation of tumor suppressor adenomatous polyposis coli (APC), causes the elevated cell proliferation rate, thus promoting the increased tumor multiplicity and tumor size, and reducing mice survival [75].

#### 2.2.5. TMED Family of p24 Proteins

TMED family members play a relevant role in cancer proliferation. TMED2 promotes ovarian cancer cell proliferation through activating the IGF2/IGF1R/PI3K/AKT signaling pathway, as described in a previous section [52]. TMED3 enhances breast cancer cell proliferation by stimulating the Wnt/β-catenin cascade, which causes an increased expression of relevant cell cycle proteins, including CDK4, c-myc and cyclinD1 [55]. Similarly, the TMED3-mediated activation of AKT/GSK3β/β-catenin axis enhances the expression of c-myc and cyclin D1, thus leading to in vitro NSCLC cell proliferation and in vivo xenograft tumor growth [56]. TMED10, interrupting TGF-β receptor complex formation, reduces the breast cancer xenograft tumor growth through negatively modulating TGF-β-induced pro-oncogenic signaling [59]. TGF-β signaling is not known to promote or inhibit cancer progression context-dependently. Therefore, it could be speculated that cancer cells themselves modulate TMED10 expression depending on the way in which this signaling acts on their survival or death.

#### 2.2.6. Similar Expression to FGF (Sef) (also Known as Interleukin-17 Receptor D (IL-17RD))

Sef inhibitory effects on cancer cell proliferation are observed in several cancer types. The ectopic expression of Sef suppresses breast carcinoma cell proliferation, whereas the inhibition of endogenous Sef expression promotes FGF- and EGF-dependent proliferation of cervical carcinoma cells [45]. Sef inhibits in vitro prostate cancer cell proliferation and in vivo prostate xenograft tumor growth by blocking FGF-induced ERK signaling, as previously described in this review [41,42]. In the same line of evidence, Sef impairs FGF2-induced MAPK/ERK signaling activation in endometrial cancer cells, thus inhibiting their growth and proliferation [46]. In addition, Sef expression is stimulated by FGF2-induced MAPK/ERK signaling, thus indicating the existence of a Sef-mediated negative feedback loop that regulates FGF cascade in endometrial cancer cells [46].

#### 2.2.7. UbiA Prenyltransferase Domain-Containing Protein 1 (UBIAD1)

UBIAD1 is a prenyltransferase localized in the GC and the ER and involved in the biosynthesis of vitamin K2 and coenzyme Q10 using geranylgeranyl diphosphate, which is necessary during the transport of this protein from the ER to the GC. UBIAD1 is down-regulated in bladder and prostate carcinomas, and its reduced expression stimulates cancer cell proliferation by activating the Ras-MAPK signaling pathway [78,79]. From a molecular point of view, the transport of UBIAD1 to the GC in the presence of geranylgeranyl diphosphate causes UBIAD1 to interact with the C-terminus of H-Ras in the GC, thus increasing its retention at the GC and preventing H-Ras trafficking from the GC to plasma membrane. This action results in inhibiting the aberrant activation of the Ras-MAPK signaling cascade at the plasma membrane and, consequently, suppressing the proliferation of bladder cancer cells [80].

#### 2.2.8. Secretory Carrier-Associated Membrane Protein 3 (SCAMP3)

SCAMP3, a component of post-Golgi membranes, functions as a protein carrier involved in subcellular protein transport. It regulates the trafficking of receptors, including EGFR, by generating multivesicular bodies in an EGF-dependent manner, thereby modulating EGFR endosomal sorting and degradation [143]. SCAMP3 is highly expressed in breast cancer, HCC and glioma [82,144,145]. SCAMP3 looks like a pro-oncogenic protein whose increased expression significantly correlates with vascular invasion and tumor stage in HCC [145] and with tumor size and poor overall survival in glioma [82]. SCAMP3 inhibits EGFR degradation and promotes its recycling, thus stimulating the EGFR signaling [81]. In addition, it is involved in the process of mTORC1 signaling activation [83]. In line with these findings, Li C and collaborators demonstrate that SCAMP3 promotes glioma proliferation through enhancing EGFR and mTORC1 signaling [82].

#### 2.2.9. Golgi Calcium Pump Secretory Pathway Calcium ATPase 1 (SPCA1)

SPCA1 is a calcium pump localized at the GC involved in regulating GC luminal calcium levels. Interestingly, the SPCA1 expression level is significantly increased in basal-like breast cancer subtypes compared to the other molecular breast cancer subtypes. In addition, its level is also elevated with increasing tumor grade, thus suggesting a pro-tumoral role for SPCA1 [84]. SPCA1-decreased expression causes the alteration of trans-Golgi Ca^2+^ content, which results in the altered regulation of calcium-dependent enzymes within the secretory pathway (such as proprotein convertases), the dysregulation of proteins sorting to the plasma membrane and the alteration of the entire GC structure [146]. In basal-like breast cancer cells, SPCA1 silencing inhibits the processing of IGF1R, a substrate of proprotein convertases involved in breast cancer progression [147]. Indeed, the IGF1/IGF1R system stimulates the FAK signal transduction pathway activation, which, in turn, regulates the nuclear accumulation of YAP (yes-associated protein/yes-related protein) and the expression of its target genes, thus inducing breast cancer cell proliferation [147]. SPCA1 knockdown induces the significant accumulation of inactive pro-form of IGF1R at the TGN and reduces the production of functional IGF1Rβ at the plasma membrane, thus resulting in the inhibition of breast cancer cell proliferation [84].

#### 2.2.10. Protein Kinase D (PKD) Family

PKD1 overexpression promotes breast cancer cell proliferation through accelerating G0/G1 to S phase transition in the cell cycle. In addition, it reduces the serum- and anchorage-dependence for proliferation and survival and enhances in vivo breast tumor growth. The pro-growth/survival effects of PKD1 on breast cancer cells are specifically mediated through activating a MEK/ERK-dependent signaling pathway and are totally independent of the PI3K/AKT cascade [92]. As previously described in this review, invasive tumor cells express low levels of PKD1; in addition, its overexpression impairs breast cancer cell invasion [140] and promotes breast cancer cell proliferation [92]. These findings suggest that PKD1 could be a switch that, according to its expression level, would lead either to cell proliferation (high expression levels) or to invasion (low expression levels). 

On the contrary, PKD1 inhibits prostate cancer cell proliferation [91]. Here, PKD1 interacts with β3-integrin, thus stimulating the MEK/ERK signaling cascade, which causes an increased expression, secretion and activation of MMP-2 and MMP-9. In turn, MMP-2 and MMP-9 promote the proteolytic cleavage of the extracellular domain of E-cadherin, which results in changes in cell adhesion, signaling, anoikis and apoptosis [148,149]. Therefore, PKD1-induced E-cadherin shedding suppresses prostate cancer cell proliferation as well as colony formation [91]. These findings highlight the importance of the cellular context that allows the engagement of a protein with specific partners into peculiar protein complexes regulating pro- or anti-proliferative signaling pathways. 

Liou GY and collaborators show that PKD1 contributes to very early events in pancreatic cancer development. They identify an oncogenic Kras mutation (Kras^G12D^ or Kras^G12V^)-induced signaling cascade involving PKD1 that plays a role in promoting pancreatic carcinogenesis [93]. The oncogenic Kras mutations alter mitochondrial metabolism, thus leading to increased levels of reactive oxygen species (ROS), which, in turn, trigger PKD1. ROS-activated PKD1 stimulates transcription factors NF-κB1 and NF-κB2, which up-regulate the expression of EGFR and its ligands TGFα and EGF, thus inducing EGFR/Kras^WT^ signaling cascade and, consequently, pancreatic cancer proliferation and malignant progression [93]. In addition, PKD1 also participates in the signaling events downstream of the TGFα/EGFR axis [94]. Herein, TGFα/EGFR induces the activation of endogenous Kras, which, in turn, stimulates PKD1. Active PKD1 inhibits the expression of Cbl and Sel1l genes, both suppressors of Notch signaling, and increases the expression of Adam10, Adam17 and MMP7, all proteinases that mediate Notch activation. These data indicate that active PKD1 acts through the Notch signaling pathway to mediate pancreatic malignant transformation [94].

PKD2 promotes the proliferation of several cancer types, including glioblastoma, colorectal, pancreatic, breast, prostate and gastric cancers [100,101,102,150,151,152], through regulating multiple signaling pathways. The PKD2 pro-proliferative function is mediated by inducing the PI3K/AKT/mTOR signaling pathway via GOLPH3 [101], an oncogene that stimulates cancer cell growth by regulating this signaling cascade [153]. PKD2 modulates GOLPH3 through two mechanisms of action: on the one side, PKD2 phosphorylates and activates phosphatidylinositol-4 kinase IIIβ (PI4KIIIβ) at the GC, which, in turn, phosphorylates phosphatidyl inositol generating PtdIns(4)P, which is required for GC localization of GOLPH3; on the other side, PDK2 positively regulates the GOLPH3 protein level. These PKD2-mediated actions cause GOLPH3-induced activation of the PI3K/AKT/mTOR signaling pathway, thus promoting cancer cell proliferation [101]. PKD2 promotes colorectal cancer cell proliferation and survival by triggering the AKT, ERK and NF-κB signaling pathways [102]. In order to identify the PKD2-regulated signaling pathways that mediate its oncogenic functions in breast cancer, Liu Y and collaborators performed an integrated phosphoproteome, transcriptome and interactome analysis [154]. Their findings show that ELAVL1 plays an important role in mediating the oncogenic functions of PKD2. ELAVL1 silencing impairs in vitro and in vivo breast cancer cell proliferation as PKD2 silencing does. PKD2 interacts with ELAVL1, and PKD2 silencing leads to ELAVL1 translocation from the cytoplasm to the nucleus without significantly affecting ELAVL1 expression [154]. 

PKD3 is highly expressed in prostate cancer and contributes to prostate cancer cell growth and survival [105]. From a molecular point of view, PKCε promotes the activation and nuclear localization of PKD3. Active PKD3 stimulates PI3K and p38, which, in turn, trigger AKT. In addition, active PKD3 promotes the phosphorylation and activation of ERK1/2. The PKD3-mediated induction of AKT and ERK1/2 signaling pathways results in prostate cancer cell proliferation by accelerating the G0/G1 to S phase transition and survival [105]. PKD3 is also elevated in triple-negative breast cancer (TNBC), where it promotes TNBC cells spreading and proliferation by triggering the mTORC1-S6K1 signaling pathway [106]. PKD3 at the GC is required for the activation of mTORC1 at endolysosomal membranes and for endosome maturation and trafficking, thus providing a molecular connection between GC-mediated protein synthesis and sorting and endolysosomal compartments-mediated catabolic processes to enhance proliferative mTORC1-S6K1 signaling [106].

### 2.3. GC-Centered Signaling Pathways that Regulate Survival and Apoptosis

#### 2.3.1. CLIPR-59 (Cytoplasmic Linker Protein 170-Related 59 kDa Protein)

CLIPR-59 is a TGN-localized protein, also associated with the plasma membrane and lipid rafts, which regulates membrane trafficking, microtubule dynamics, AKT cellular compartmentalization and TNFα-induced apoptosis. CLIPR-59 low expression is detected in glioblastoma and high-grade glioma compared to low-grade glioma and normal tissues, and it is associated with glioma highly aggressive phenotype [85], thus suggesting that CLIPR-59 serves as a tumor suppressor. CLIPR-59 plays a role in glioblastoma resistance to TNFα-mediated apoptosis [85]. In the context of the TNFα signaling pathway, CLIPR-59 is an adaptor protein for TNFR1, which binds TNFR1 in resting cells. Upon TNFα stimulation, several proteins interacting with TNFR1, including receptor-interacting protein 1 (RIP1) and TNF receptor-associated death domain protein (TRADD), are recruited to the receptor to form a complex, called Complex-I, which facilitates the ubiquitination of RIP1. CLIPR-59 subsequently interacts with the de-ubiquitinating enzyme CYLD, thus scaffolding CYLD into the complex containing RIP1, which leads to the de-ubiquitination of RIP1 at lysine 63. The de-ubiquitination of RIP1 induces the recruitment of Caspase-8 and Fas-associated protein with death domain (FADD) to generate another complex, named Complex-II. In Complex-II, Caspase-8 is activated and promotes apoptosis [155]. Another player in the TNFα-mediated apoptosis is Human Speedy A1 (Spy1), a member of the Speedy/RINGO family, which promotes cell survival, prevents apoptosis and inhibits checkpoint activation in response to DNA damage [156]. In gliomas, Spy1 is highly expressed; therefore, its expression negatively correlates to CLIPR-59 expression. CLIPR-59 and Spy1 interact, and this interaction suppresses the association of CLIPR-59 and CYLD during TNFα signaling, impairing the RIP-1 lysine-63-dependent de-ubiquitination and, consequently, the activation of the apoptosis process [85]. Therefore, this mechanism of action suggests that the CLIPR-59/Spy1 interaction, CLIPR-59 decreased expression, and Spy1 increased expression represent the molecular bases underlying the glioblastoma resistance to TNFα-induced apoptosis. 

#### 2.3.2. Ras

Ras is present and functional not only at the plasma membrane but also at endomembranes, such as ER, endosomes and GC [157]. Based on its different cellular localization, Ras is subject to site-specific regulation by distinct exchange factors, engages alternative effector pathways and switches on diverse genetic programs, which results in the differential potential to drive carcinogenesis. Casar and collaborators showed that in breast cancer cells, TGF-β induces the activation of a Ras pool localized at cis Golgi, which leads to cancer cell apoptosis [86]. The presence of activated Ras, both endogenous and ectopic, at the cis Golgi triggers the RAL GEFs effector pathway, which activates RAL GTPases responsible for the induction of JNK and p38 pro-apoptotic MAPKs [158,159] and the inhibition of NF-κB survival factor [160]. These actions result in antagonizing ERK activation, thus stimulating an apoptotic response and preventing the malignant transformation induced by oncogenic signals coming from other subcellular localizations or from other oncogenes (including v-Src, v-Sis and ERB2) [86]. From a molecular point of view, activated Ras at cis Golgi induces the expression of Protein Tyrosine Phosphatase receptor kappa (PTPRκ), which binds to and dephosphorylates c-Raf, thus reducing Ras-stimulated c-Raf activation and, consequently, the phosphorylation levels of MEK and ERK. This cascade of events culminates in apoptosis induction [86]. Interestingly, the GC-localized oncogenic HRasV12 is deficient for inducing melanoma in a zebrafish animal model, while it induces melanoma when its signals emanate from the plasma membrane, thus emphasizing the role played by the Ras pool signals emanating from the GC in antagonizing tumor development [86].

#### 2.3.3. TMED Family of p24 Proteins

A recent study demonstrated that TMED3 stimulates in vitro and in vivo survival and proliferation and suppresses apoptosis of chordoma cancer cells, thus serving as a positive cancer regulator [57]. Although the molecular mechanisms by which TMED3 promotes chordoma progression are not fully understood, evidence shows that TMED3 plays these functions through activating PI3K/AKT signaling and inhibiting apoptosis and MAPK9/JNK2 signaling pathways [57]. Likewise, TMED10 negatively modulates PKCδ-mediated apoptosis in prostate cancer cells. TMED10 binds to and retains PKCδ to the perinuclear region, thus impairing its translocation to the plasma membrane and activation in response to phorbol esters. Similarly, TMED10 retains PKCδ to the perinuclear region, thus impairing its translocation to the nucleus and activation in response to chemotherapeutic drugs. The TMED10-mediated limitation of PKCδ availability results in the suppression of activation of PKCδ downstream effectors ROCK and JNK and, consequently, in the inhibition of stimuli-induced apoptosis in prostate cancer cells [60].

#### 2.3.4. Protein Kinase D (PKD) Family

PKD1 and PKD2 play anti-apoptotic and pro-survival roles in response to the apoptotic agent PMA, a phorbol ester, in LNCaP prostate cancer cells [104]. PMA triggers PKCδ and PKCε, which drives a rapid activation of endogenous PKD proteins. The PMA-mediated activation of PKD1 or PKD2 results in a dual action: on the one hand, it induces the transcriptional activities of ERK1/2 and NF-κB signaling pathways involved in LNCaP cell survival and, on the other hand, reduces SAKP/JNK activity, a pro-apoptotic signal in LNCaP cells. Therefore, the PMA-mediated activation of PKD proteins induces pro-survival signals that suppress PMA-induced apoptotic response. Then, the PMA prolonged treatment stimulates the slow and progressive down-regulation of endogenous PKD1, which is mediated by the PKC-dependent ubiquitin–proteasome degradation pathway, thus facilitating PMA-induced apoptosis [104].

### 2.4. GC-Centered Signaling Pathways that Regulate Autophagy

Autophagy is a critical regulator of cellular homeostasis, and autophagic dysfunction is associated with several human diseases, including cancer. Although autophagy has complex and context-dependent roles in cancer, its involvement in tumorigenesis and cancer hallmarks is now recognized [161]. Therefore, the current knowledge on the GC-centered signaling pathways that regulate autophagy in cancer is reviewed.

#### 2.4.1. TMED Family of p24 Proteins

TMED10 contributes positively to papillary thyroid cancer cell proliferation by inhibiting autophagy through suppressing the adenosine monophosphate (AMP)-activated protein kinase (AMPK)/mTOR pathway [61].

#### 2.4.2. VPS53

VPS53 is one of the subunits of the Golgi-associated retrograde protein (GARP) complexes. The GARP complex is involved in intracellular cholesterol transport and sphingolipid homeostasis by mediating retrograde trafficking from endosomes to the GC. The functional dysregulation of the GARP complex causes the alteration of sphingolipid and sterol homeostasis and, in turn, the accumulation of sphingolipid synthesis intermediates in the lysosomes, thus leading to lysosomal dysfunction [162]. In addition, the GARP complex participates in recycling and the stabilization of the GC glycosylation machinery. The depletion of GARP subunits, including VPS53, results in glycosylation defects and a decreased level of GC-resident proteins and enzymes, thus leading to functionally aberrant glycoproteins [163]. VPS53 expression is strongly reduced and positively correlates with the expression of the autophagy-related gene *Beclin1* in colorectal cancer tissue. VPS53 overexpression induces the autophagy signaling pathway, as exemplified by the increased expression of autophagy-related proteins, including LC3BII and Beclin 1, thus promoting autophagy and apoptosis and, in turn, impairing the proliferation, migration and invasion of colorectal cancer cells [87]. Although the molecular mechanisms underlying the VPS53-mediated regulation of the autophagy signaling pathway have not been revealed yet, these findings suggest that VPS53 is a tumor suppressor of colorectal cancer progression.

### 2.5. GC-Centered Signaling Pathways that Regulate Angiogenesis

#### 2.5.1. RKTG (Raf Kinase Trapping to Golgi)/PAQR3

In clear-cell renal cell carcinoma (ccRCC), RKTG/PAQR3 expression is reduced and inversely correlates with VEGF expression. RKTG/PAQR3 impairs the angiogenesis and tumorigenesis of ccRCC by two molecular mechanisms [76]. RKTG/PAQR3, through suppressing Ras/Raf/MEK/ERK signaling cascade, impairs the formation of hypoxia-inducible factor-1α (HIF-1α)/p300 complex, which results in the inhibition of the transactivation activity of HIF-1α and, in turn, of *VEGF* transcription, thereby reducing hypoxia-induced VEGF production. In addition, RKTG/PAQR3 suppresses the VEGF-mediated activation of ERK signaling, which causes the inhibition of endothelial cell proliferation, migration and tube formation. Therefore, RKTG/PAQR3 down-regulation in ccRCC promotes HIF-1α-mediated VEGF autocrine function and VEGF-induced angiogenesis [76].

#### 2.5.2. Protein Kinase D (PKD) Family

PKD2 plays a role in promoting angiogenesis in low oxygen conditions through two molecular mechanisms. Hypoxia induces the expression of HIF-1α transcription factor, which activates the expression of *VEGF*, thus inducing the tumor angiogenesis. PKD2 mediates the hypoxia-induced accumulation of HIF-1α. In addition, PKD2 stimulates the phosphorylation and proteasomal degradation of IkBα, thus triggering the NF-κB signaling pathway that culminates in the expression of angiogenic factors including VEGFα, thus promoting tumor angiogenesis and growth [103].

Another mechanism by which PKD proteins promote tumor angiogenesis consists of remodeling the tumor microenvironment. PKD2/3 promotes prostate cancer angiogenesis through regulating mast cell recruitment and microvessel density in the tumor microenvironment [107]. In prostate cancer cells, PKD2/3 activation triggers the ERK1/2 and NF-κB signaling pathways. Consequently, AP-1, the key transcriptional factor of ERK1/2 signaling, and NF-κB bind to the promoters of specific chemokines, including SCF, CCL5 and CCL11, thus resulting in their increased expression and secretion. These secreted chemokines, in turn, promote the recruitment of mast cells in the tumor microenvironment and the expression of mast cell angiogenic factors such as VEGF, TNFα, IL-6, IL-8 and FGF-2, which induce the tumor angiogenesis [107]. 

### 2.6. GC-Centered Signaling Pathways that Regulate Cancer Stemness

#### 2.6.1. TMED Family of p24 Proteins

As previously described in this review, TMED3 positively modulates the WNT-TCF signaling cascade in colon cancer, thus suppressing cancer metastases. WNT-TCF signaling is involved in several aspects of tumorigenesis, including the promotion and maintenance of the cancer stem cells (CSCs) population, which may underlie metastases. Based on this consideration, Duquet and collaborators [53] show that TMED3 silencing causes a significant reduction in colon CSCs clonogenicity, thus suggesting a role for TMED3 in cancer stemness. In addition, the TMED3-silenced spheroids show single cell protruding and spreading, thus indicating the invasive behavior of the cell population. Moreover, TMED3 silencing induces the down-regulation of WNT-TCF target genes involved in colon and colon cancer stemness, including ASCL2 and LGR5. On the other hand, the TMED3 silencing-induced inhibition of WNT-TCF signaling results in the enhanced activity of HH-GLI signaling, which stimulates the expression of CSC-related factors, including NANOG, SOX2, OCT4 and KLF4. These findings support the hypothesis that the signaling pathway switch promotes a change in tumor stem cell identity from tissue-specific phenotype to more metastatic states, which support metastases formation [53].

#### 2.6.2. Protein Kinase D (PKD) Family

PKD1 is a key regulator of the stemness of breast cancer stem cells (BCSCs). PKD1 activates the GSK3/β-catenin signaling pathway by enhancing the inhibitory phosphorylation of GSK3α/GSK3β, which causes the increased level of β-catenin, thus promoting the enrichment of the BCSCs population [95]. The role of PKD1 in promoting breast cancer stemness is further corroborated by Jiang Y and collaborators, who demonstrate that in estrogen receptor-positive breast cancer lysophosphatidic acid (LPA) induces PKD1 activation, which, in turn, stimulates the MAPK-ERK1/2 signaling pathway, thus resulting in the transcription of stemness-associated genes, including Notch1, ALDH1, CD36, CD44 and KLF4 [96].

### 2.7. GC-Centered Signaling Pathways that Regulate Cancer Resistance to Therapies

#### 2.7.1. TMED Family of p24 Proteins

TMED3 promotes the resistance of NSCLC cells to cisplatin chemotherapy through activating the AKT/GSK3β/β-catenin axis, as previously described [56].

#### 2.7.2. Protein Kinase D (PKD) Family

PKD1 expression is associated with breast cancer drug-resistance properties. PKD1 stimulates breast cancer drug resistance by promoting breast cancer stemness through the activation of the GSK3/β-catenin signaling pathway, as previously described [95]. 

### 2.8. GC-Centered Signaling Pathways that Reprogram Cancer Metabolism

#### Protein Kinase D (PKD) Family

PKD1 regulates the glycolytic metabolism of cancer cells in hypoxia conditions [97]. In oral squamous cell carcinoma cells, hypoxia induces the expression and activation of PKD1, which, in turn, activates the p38 MAPK signaling cascade, thus stimulating the expression and activation of HIF-1α and, consequently, the metabolic switch of cancer cells. Therefore, PKD1 stimulates glucose consumption and L-lactate production, which results in the increased synthesis of lipids and nucleotides and the stimulation of growth and invasion [97]. Similarly, PKD1 promotes pancreatic cancer tumorigenesis, chemoresistance and progression through reprogramming cancer cell glucose metabolism [98]. Herein, PKD1 activates mTORC1, which, in turn, phosphorylates its downstream effectors S6K and 4EBP1, which phosphorylate/activate downstream proteins involved in initiation and elongation, thus stimulating the expression of proteins involved in metabolic switch and glucose metabolism, including glucose transporter-1 (GLUT1) and HIF-1α [98].

### 2.9. GC-Centered Signaling Pathways that Regulate Chronic Inflammation

#### 2.9.1. Golgi Membrane Protein 1 (GOLM1)/Golgi Protein 73 (GP73)/Golgi Phosphoprotein 2 (GOLPH2)

Although GOLM1 has been recognized as an oncogene that promotes several malignancies, a recent study provides evidence that this protein acts as a tumor suppressor in colitis-associated colorectal cancer [40]. In intestinal epithelial cells, GOLM1 interacts with cleaved NOTCH2 (N2ICD), and retains it in the cytoplasm, thus impairing its nuclear translocation and the activation of its target genes. Decreased GOLM1 expression leads to the enhanced activation of the Notch2 signaling pathway, which alters lineage specification and differentiation of intestinal epithelial cells, thus leading to sustained mucosal inflammation, colitis-induced epithelial damage and, consequently, colon cancer development [40].

#### 2.9.2. Similar Expression to FGF (Sef) (also Known as INTERLEUKIN-17 receptor D (IL-17RD))

Previous studies demonstrate that Sef is a negative regulator not only of mitogenic signaling, as previously discussed in this review but also of inflammation signaling. Indeed, it suppresses both NF-κB and interferon regulatory factor (IRF) signaling pathways initiated by pro-inflammatory cytokine and Toll-like receptors (TLR), thus causing the attenuation of pro-inflammatory gene expression. Vice versa, Sef deficiency results in enhanced NF-κB and IRF cascade activation and the up-regulation of pro-inflammatory cytokines [47,48]. Based on these findings, Girondel and collaborators investigated the role of Sef in colitis-associated tumorigenesis, which is driven by dysregulated mitogenic signaling and chronic inflammation [49]. In Sef knockout mice induced to develop colitis-associated colorectal cancer, the Sef loss stimulates colon tumorigenesis by promoting the activation of TLR and IL-17 signaling, which enhances STAT3 tyrosine phosphorylation, thus leading to the expression of pro-inflammatory cytokines, including IL-17A and IL-6. Thus, the down-regulation of Sef expression favors the creation of an inflammatory tumor microenvironment, characterized by a higher colitis score, increased immune cell infiltration and an increase in circulating pro-inflammatory cytokines, conducive to tumor development. These findings demonstrate the role of Sef in impairing tumorigenesis by limiting the extent and duration of inflammation [49].

### 2.10. GC-Centered Signaling Pathways that Regulate Cancer Genomic Instability

#### Similar Expression to FGF (Sef) (also Known as Interleukin-17 Receptor D (IL-17RD))

Sef plays a role in inhibiting the Ras-mediated polyploidization of the cells [50]. In colorectal tumors and colon cancer cell lines, the Ras oncogenic activation reduces Sef expression very early during the oncogenesis, thus driving the aberrant nuclear accumulation of phosphorylated MEK1/2 and ERK1/2, which leads to ERK1/2 signaling hyperactivation, which, in turn, causes cell-cycle dysregulation, increased cell proliferation, polyploidization and neoplastic transformation. Sef re-expression in Ras-transformed cells is sufficient to rescue the normal cytoplasmic localization of phosphorylated MEK1/2, attenuate the activating phosphorylation of ERK1/2, and reverse the transformed morphological phenotype and prevent Ras-mediated genomic instability [50].

## 3. Conclusions

The classical functions of GC identify this organelle as the subcellular compartment that plays a central role in the orchestrating protein and lipid glycosylation and their trafficking to the final destination. In addition to these functions, accumulating evidence shows that GC is the hub of signaling networks that contribute to the regulation of a range of cellular processes, including mitosis, migration, DNA repair, stress responses, autophagy, apoptosis and inflammation, whose dysregulation leads to the pathogenesis of several diseases, including cancer. This review focuses on the GC-centered signaling pathways whose alteration promotes cancer progression. Several GC scaffold proteins and GC-localized signaling molecules are involved in more than one signaling cascade and contribute to promoting different cancer hallmarks (Table 1). Multiple molecular mechanisms are evoked by GC-localized proteins to affect the signaling pathways involved in carcinogenesis (Figure 2). The GC-localized proteins may have multiple roles in signaling. They can serve as scaffolds for a complex signaling formation (Figure 2A). They can act as an anchor that sequesters either a positive regulator of a complex signaling, disrupting the signaling complex formation (Figure 2B) or an inhibitory regulator of the complex, leading to the activation of the signaling cascade (Figure 2C). Specific GC proteins can interact with and sequester signaling proteins to GC, impairing their activity in the signaling cascade (Figure 2D), or can directly bind to and phosphorylate signaling proteins, modulating the activity of downstream effectors (Figure 2E), or modulate the stability of signaling proteins through promoting the protein ubiquitination and degradation (Figure 2F). Finally, the GC can further control signaling by modulating the processing, intracellular sorting and secretion of signaling proteins/molecules and cancer-related proteins (Figure 2G). In addition, the function of some proteins in carcinogenesis is context-dependent. As tackled in this review, GM130, PKD1, GOLM1, TMED3 and TMED10 play pro-tumoral or anti-tumoral functions depending on the interaction with specific partners engaged in unique cellular/tumoral context. 

The role played by GC in carcinogenesis makes this organelle and its cancer-involved proteins potential candidates for anti-cancer therapy. The ARF1 targeting via siRNA or agents disrupting its function sensitizes TNBC cells to antitumor drugs and EGFR tyrosine kinase inhibitors [164,165]. GOLM1 induces tumor growth and metastasis and leads to poor survival in patients. Its targeting represents a new therapeutic avenue in cancer treatment. However, its intracellular localization and the lack of domains that could possibly be interfered with small molecules make it very difficult to target GOLM1. Recently, the natural product epigallocatechin gallate has been identified as the first compound able to reduce GOLM1 expression, thus leading to the inhibition of TNBC cell migration [166]. Some GC-localized proteins act as tumor suppressors whose expression is strongly reduced in cancer. In this case, the therapeutic approaches are aimed at rescuing their expression. Therapeutic ultrasound waves (TUS) are a non-viral approach for the non-invasive delivery of genes into cells and tissues approved for clinical application. TUS-mediated Sef delivery into prostate tumors inoculated in mice suppresses tumor growth and angiogenesis, thus demonstrating the efficacy of this approach for the treatment of carcinomas where the expression of tumor suppressors is down-regulated [167]. Although these preclinical studies demonstrate the therapeutic efficacy of the strategies based on GC targeting, their translation to clinical trials demands further studies. 

In summary, GC plays a relevant role in regulating cancer-involved signaling pathways. Although many other GC-localized proteins, in addition to those addressed in this review, are involved in carcinogenesis, the underlying signaling pathways remain widely unrevealed. Further studies will allow the unraveling of the molecular mechanisms through which these proteins act and will provide additional potential candidates for developing novel therapeutic purposes.

## Figures and Tables

**Figure 1 cells-11-01990-f001:**
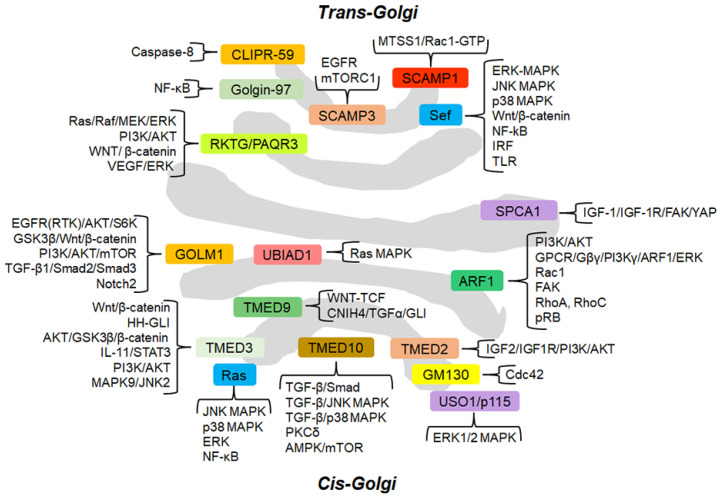
Schematic representation of the GC and GC-localized proteins (approximately in their localization) involved in cancer-related signaling pathways. The signaling cascades modulated by GC-proteins are grouped and briefly described. For the details of the molecular mechanisms through which the GC-localized proteins regulate the signaling pathways, the reader is referred to the text.

**Figure 2 cells-11-01990-f002:**
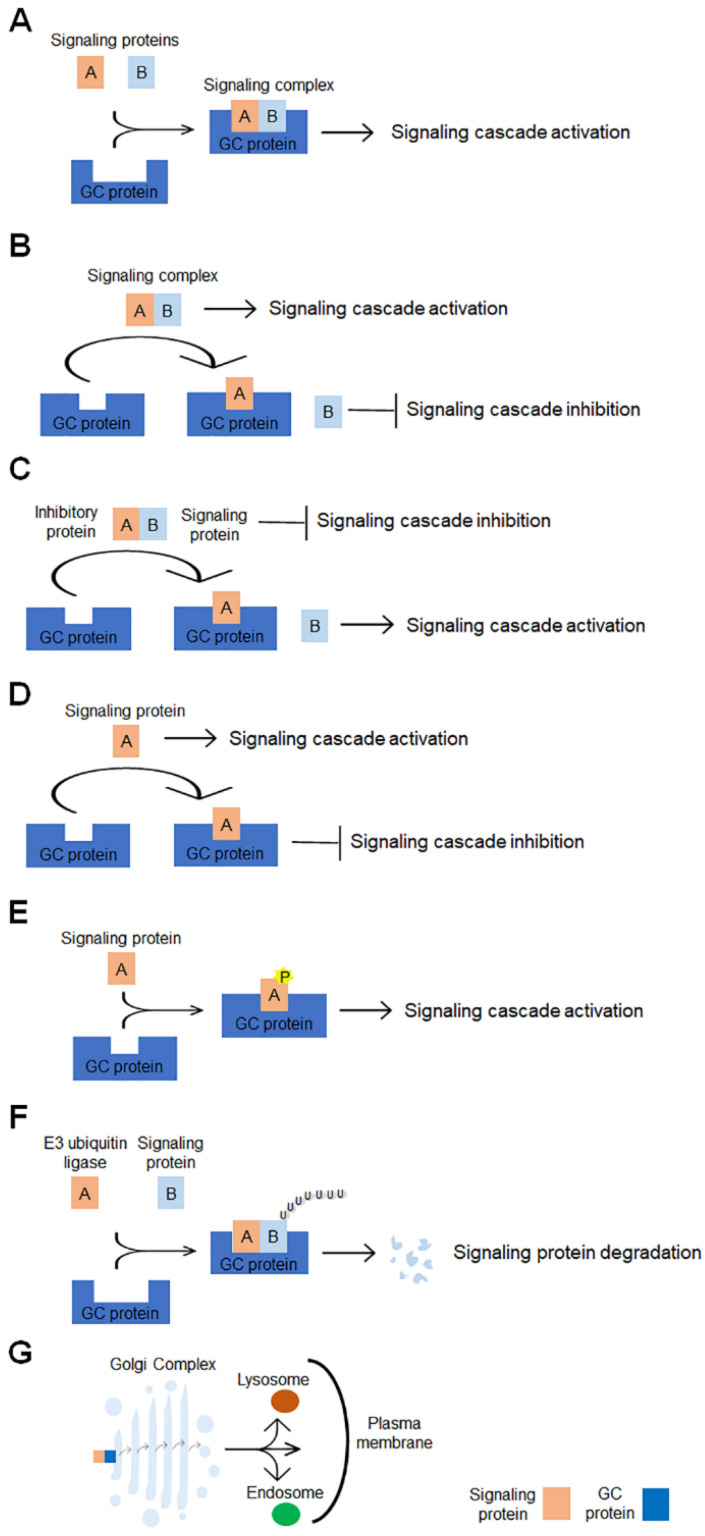
Schematic representation of molecular mechanisms underlying the GC-mediated regulation of signaling pathways involved in tumorigenesis. The GC-localized proteins can modulate the signaling pathways via multiple molecular mechanisms: (**A**) they can promote the complex signaling formation, (**B**) they can disrupt the complex signaling formation, (**C**) they can sequester inhibitory proteins, thus preventing their interaction with signaling molecules, (**D**) they can bind to signaling molecules, thus impairing the signaling cascade activation, (**E**) they can bind to, phosphorylate and activate signaling molecules, (**F**) they can enhance the signaling protein degradation and (**G**) they can promote the signaling molecules processing, intracellular sorting and secretion.

**Table 1 cells-11-01990-t001:** GC protein-regulated pathways implicated in cancer progression.

Protein	Expression Change in Cancer	Function in Cancer	Molecular and Signaling Pathways Regulated	Cancer Hallmarks	References
ARF1	Increased	Oncogene	↑PI3K/AKT	↑Migration	[19,20]
			↑Rac1	↑Invasion	[21]
			↑FAK		[22]
			↑RhoA, RhoC		[23]
			↑GPCR-Gβγ-PI3Kγ-ARF1-ERK		[24]

			↑PI3K/AKT		
			↑pRb phosphorylation	↑Proliferation	[19,20]
					[25]
GM130	Increased	Oncogene	↑Snail transcription	↑Migration ↑Invasion	[26]

				↓Migration ↓Invasion	
	Decreased	Tumor suppressor	↑GC-localized Cdc42		[27,28]
GOLM1	Increased	Oncogene	↑CREB expression	↑Migration ↑Invasion	[29]
			↑MMP2 trafficking and transcription	↑Metastasis	[30]
			↑EGFR(RTK)/AKT/S6K		
			↑GSK3β		[31]
			↑PI3K/AKT/mTOR		[32]
			↑TGF-β1/Smad2/Smad3		[33,34]
			↑Wnt/β-catenin		[35,36]
			↓p53 stability		[37]
					[38]
			↑PI3K/AKT/mTOR	↑Proliferation	
			↑Wnt/β-catenin	↑Tumor growth	[31,32,33,34,39]
					[37]
			↓Notch2	↓ Cancer inflammation	
	Decreased	Tumor suppressor			[40]
Sef	Decreased	Tumor suppressor	↓ERK MAPK	↓Migration ↓Invasion	[41,42]
			↓JNK MAPK	↓Metastasis	[43]
			↓p38 MAPK		[43]
			↓Wnt/β-catenin		[44]
				↓Proliferation	
			↓ERK MAPK	↓Tumor growth	[41,42,45,46]

				↓Cancer inflammation	
			↓NF-kB		[47]
			↓IRF		[48]
			↓TLR	↓Polyploidization	[48,49]
				↓Genomic instability	
			↓ERK1/2 MAPK		[50]
Golgin-97	Decreased	Tumor suppressor	↓NF-κB	↓Migration	[51]
				↓Invasion	
TMED2	Increased	Oncogene	↑IGF2/IGF1R/PI3K/AKT	↑Migration	[52]
				↑Invasion	
				↑Proliferation	
TMED3	Increased	Metastasis suppressor	↑WNT-TCF	↓Metastasis ↓Embryonic-like metastatic CSCs population	[53]
			↓HH-GLI signaling		
				↑Migration	
				↑Invasion	
		Metastasis promoter		↑Metastasis	
			↑IL-11/STAT3		[54]
				↑Migration	
		Oncogene		↑Invasion	
				↑Proliferation	
			↑Wnt/β-catenin	↑Tumor growth	[55]
			↑AKT/GSK3β/β-catenin		[56]
				↑Survival	
		Oncogene		↑Proliferation	
				↑Tumor growth	
			↑PI3K/AKT	↓Apoptosis	[57]
			↓MAPK9/JNK2		
			↓Apoptosis signaling	↑Chemoresistance	


			↑AKT/GSK3β/β-catenin axis		[56]
TMED9	Increased	Metastasis promoter	↓WNT-TCF	↑Migration	[58]
			↑CNIH4/TGFα/GLI	↑Metastasis	
TMED10	Increased	Tumor suppressor	↓TGF-β/Smad	↓Migration	[59]
			↓ TGF-β/JNK MAPK	↓Tumor growth	
			↓ TGF-β/p38 MAPK		

		Oncogene	↓PKCδ	↓Apoptosis	[60]

			↓AMPK/mTOR	↑Proliferation	[61]
SCAMP1		Tumor suppressor	↑MTSS1/Rac1-GTP axis	↓Migration ↓Invasion	[62]
RKTG/PAQR3	Decreased	Tumor suppressor	↓Ras/Raf/MEK/ERK	↓Migration ↓Invasion	[63,64,65,66]
			↓PI3K/AKT	↓Metastasis	[67,68,69]
			↓Twist1 stability		[70]
				↓Proliferation	
			↓Ras/Raf/MEK/ERK	↓Tumor growth	[63,64,66,68,71,72,73]
					[68,69,73,74]
			↓PI3K/AKT		[75]
			↓WNT/ β-catenin		
				↓Angiogenesis	[76]
			↓Ras/Raf/MEK/ERK	↓Endothelial cells proliferation, migration and tube formation	
			↓VEGF/ERK axis		



USO1/p115	Increased	Oncogene	↑ERK1/2 MAPK	↑Proliferation	[77]
UBIAD1	Decreased	Tumor suppressor	↓Ras MAPK	↓Proliferation	[78,79,80]
SCAMP3	Increased	Oncogene	↑EGFR signaling	↑Proliferation	[81,82]
			↑mTORC1 signaling		[82,83]
SPCA1	Increased	Oncogene	↑IGF-1/IGF-1R/FAK/YAP	↑Proliferation	[84]
CLIPR-59	Decreased	Tumor suppressor	↑Caspase-8 activation	↑TNFα-mediated apoptosis	[85]
GC-localized Ras			↑JNK MAPK	↑Apoptosis	[86]
			↑p38 MAPK		
			↓NF-κB		
			↓ERK		
VPS53	Decreased	Tumor suppressor	↑Autophagy signaling	↑Apoptosis	[87]
				↓Proliferation	
				↓Migration	
				↓Invasion	
PKD1	Decreased	Tumor suppressor	↓Wnt/β-catenin	↓Migration ↓Invasion	[88]
			↓Snail activity		[89]
			↑Interaction of E-cadherin with catenins		[90]

			↑MEK/ERK	↓Proliferation
	Increased	Tumor suppressor			[91]

			↑MEK/ERK	↑Proliferation	
	Increased	Oncogene	↑Oncogenic Kras/ROS/PKD1/NF-κB	↑Tumor growth	[92]
				↑Proliferation	[93]
			↑Notch		
				↑Malignant trasformation	
					[94]
			↑GSK3/β-catenin	↑CSCs population	
			↑LPA/PKD1/ERK	↑Cancer stemness	
					[95]
			↑GSK3/β-catenin	↑Chemotherapy resistance	[96]

				↑Metabolic reprogrammimng	[95]
			↑p38 MAPK/HIF-1α		
			↑mTORC1/pS6K, 4EBP1		
					[97]
					[98]
PKD2	Increased	Oncogene	↑PI3K/AKT/ GSK3β/β-catenin axis	↑Migration ↑Invasion	[99]
			↑NF-κB	↑Metastasis	
					[100]

			↑ PI4KIIIβ/GOLPH3/PI3K/AKT/mTOR axis	↑Proliferation	
			↑AKT, ERK, NF-κB		[101]

			↑HIF-1α accumulation		
			↑NF-κB		[102]
				↑Angiogenesis	
					[103]
PKD1/PKD2			↑ERK1/2	↑Survival	[104]
			↑NF-kB	↓Apoptosis	
			↓SAKP/JNK		
PKD3	Increased	Oncogene	↓HDAC1 expression	↑Migration ↑Invasion	[100]
				↑Metastasis	

				↑Proliferation	
			↑PI3K, p38, ERK1/2	↑Tumor growth	[105]
			↑mTORC1- S6K1		[106]
PKD2/3	Increased	Oncogene	↑ERK1/2	↑Tumor micro-environment remodeling	[107]
			↑NF-κB	↑Angiogenesis	

↑ represents the activation of signaling pathway and the induction of cellular/biological process; ↓ represents the inhibition of signaling pathway and the suppression of cellular/biological process.

## Data Availability

Not applicable.

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
