# Peer review of "Golgi Complex: A Signaling Hub in Cancer"

_cells, 2022, doi:10.3390/cells11131990_

Round 1

Reviewer 1 Report

The Golgi Complex: a signaling hub in cancer” article is a very nicely written and comprehensive review of a topical subject. As such, it makes an important contribution to the field, and I have only one suggestion for improvement. I did not see any major flaws and hopefully, they will find this comment useful while revising the manuscript. I think it would be nice to show a figure showing the Golgi complex proteins in the signaling pathway or pathway involved.

Author Response

We thank the reviewer for the critical reading of manuscript, the positive comments and the suggestion provided us to improve it.

The complexity of signaling networks and the cross-talk make very hard to compose a figure showing the Golgi Complex proteins in the signaling pathways involved. Therefore, while agreeing with the reviewer's suggestion and appreciating its relevance, we preferred to show in the new figure, called figure 1, the localization of the Golgi Complex proteins indicating the signaling pathways in which they are involved. We hope that the reviewer understands the reasons for this choice and that he/she is satisfied with the figure 1. According to the inclusion of this new figure, the figure 1 included in the previous version of the manuscript is renamed as Figure 2.

Reviewer 2 Report

I have no major comments regarding the organization and information given in this very comprehensive and well written review.

Minor comments

My advice to the authors is to briefly include the molecular mechanism for each protein or pathway mentioned in this review. This will be useful to understand whether the protein acts as positive or negative cancer regulator. I´m pretty sure that the authors will be able to briefly include these details. I think this information will enrich the manuscript.

As examples

Line 74

ARF1 is associated to the plasma membrane in some cell types and cycle off the GC to the cytosol upon specific conditions [108,109]. Might be worth for non-Golgi or membrane transport specialized readers to briefly know about the molecular mechanism that ARF uses to control membrane transport. Specially to understand whether ARF1 in normal and cancer cells is altered or not, there is a change on the protein interactor net or changes on protein levels.

Line 122

GM130 is a cis-Golgi matrix protein involved in maintenance of GC structure, in the stacking of GC cisternae, and in recruiting protein complexes involved in microtubule polimerization and plarity-based signaling. Should be interesting whether the author might include how GM130 keeps GC structure, and how different or not it is comparing normal with cancer cells. I think plarity must be replaced by polarity.

Author Response

We thank the reviewer for the critical reading of manuscript, the positive comments and the suggestion provided us to improve it.

Following the reviewer’s suggestion, we included in the text a brief description of the molecular mechanisms, if unraveled, and/or the roles played for mentioned proteins to highlight their relevance in cancer. All the changes in the text are highlighted in red. Additional details related to the proteins expression changes in cancer and the oncogenic and/or tumor suppressive function are included in Table 1 and described in the text as well. The included descriptions are the following:

Lines 73-81: Arf1 description

Lines 129-136: GM130 description

Lines 159-161: GOLM1 description

Lines 228-230: Sef description

Lines 258-263: Golgin-97 description

Lines 275-279, lines 287-288, lines 300-301, lines 342-346: TMED family members description

Lines 364-366: RKTG/PAQR3 description

Lines 487-490: USO1/p115 description

Lines 562-568: UBIAD1 description

Lines 575-577: SCAMP3 description

Lines 758-766: VPS53 description

We hope the revised manuscript addresses satisfactorily the reviewer’s advice.